# Increase in Efficiency of End Milling of Titanium Alloys Due to Tools with Multilayered Composite Nano-Structured Zr-ZrN-(Zr,Al)N and Zr-ZrN-(Zr,Cr,Al)N Coatings

**Alexey Vereschaka [1],\*****, Maksim Oganyan [1], Yuri Bublikov [2], Nikolay Sitnikov [3], Konstantin Deev [4], Vladimir Pupchin [4] and Boris Mokritskii [5]**

[1]  Department of Mechanical Engineering, Moscow State Technological University STANKIN, Moscow 127055, Russia; baks723@gmail.com
[2]  Surface modification laboratory, IKTI RAN, Moscow 127055, Russia; yubu@rambler.ru
[3]  Department of Solid State Physics and Nanosystems, National Research Nuclear University MEPhI, Moscow 115409, Russia; sitnikov_nikolay@mail.ru
[4]  Central Aerohydrodynamic Institute named after N.E. Zhukovsky, Scientific and Technical Centre, Moscow 127018, Russia; kon_stantine@mail.ru (K.D.); doctor.alchin@mail.ru (V.P.)
[5]  Department of Mechanical Engineering, Komsomolsk-na-Amure State Technical University, Komsomolsk-on-Amur 681013, Russia; boris@knastu.ru
\*  Correspondence: ecotech@rambler.ru; Tel.: +7-916-910-0413

**Abstract:** The study deals with an increase in the tool life parameter for metal-cutting tools and efficiency of end milling for titanium alloys, due to the use of tools with multilayered composite nano-structured Zr–ZrN–(Zr,Al)N and Zr–ZrN–(Zr,Cr,Al)N coatings, deposited through the technology of the filtered cathodic vacuum arc deposition (FCVAD). The studies included the microstructured investigations using SEM, the analysis of chemical composition (Energy-dispersive X-ray spectroscopy, EDXS), the determination of the value of critical failure force (with the use of scratch testing), and the measurement of the microhardness of the coatings under study. The cutting tests were conducted in end milling of titanium alloys at various cutting speeds. The mechanisms of wear and failure for end milling cutters with the coatings under study were studied in milling. The studies determined the advantages of using a tool with the coatings under study compared to an uncoated tool, as well as to tools with the commercial Ti–TiN coating and the nano-structured Ti–TiN–(Ti,Al)N coating. Adding Cr to the composition of the coating can significantly increase the hardness, while the coating retains sufficient ductility and brittle fracture resistance, which allows for a best result when milling titanium alloys.

**Keywords:** multilayered nano-structured coatings; wear; diffusion; end milling; titanium alloy

## 1. Introduction

Titanium alloys are widely used in various fields of modern engineering, in particular in aerospace. In approximately 57% of cases, such parts of gas turbine engines, as blades, rotors of centrifugal compressors, and monowheels are made of titanium alloys [1–3]. Due to their high strength, toughness, oxidizability, and low thermal conductivity, they are hard-to-cut materials, which leads to considerable difficulties in their machining. End milling is a widespread and efficient technique to machine titanium alloys, and it ensures high performance of cutting and good quality of machined surface. However, a further increase in the efficiency of cutting in end milling, in particular an increase in cutting speeds,

is prevented by the fact that modern tool materials have reached their upper limit in terms of thermal resistance [4].

There are a number of studies considering the effect of various wear mechanisms on cutting tools and ways to further improve the performance in end milling of titanium alloys. In particular, in [5], it was found that in end milling of a workpiece made of Ti–6Al–4V titanium alloy with an uncoated carbide tool, the prevailing wear mechanisms on the rake face were adhesive and diffusive wear, while on the flank face, they were adhesive and abrasive wear. Meanwhile, the component of the cutting force $P_y$ is dominant, and its change can be used to control the degree of tool wear. In [6], authors considered the process of end milling for Ti–6Al–4V and Ti-555 titanium alloys with a tool with monolayered TiAlN coating and revealed the differences in the wear mechanism for a tool in machining of the specified alloys. In particular, in milling of Ti-555 alloys, characterized by a fine-grained structure (with a grain size of about 1 μm), close to the β titanium structure, the abrasive wear of the tool prevailed. Meanwhile, in machining of Ti–6Al–4V alloy (with a grain size of about 10 μm), the adhesive and diffusive wear mechanisms prevailed. It was also found that in comparison with Ti-555 alloy, Ti–6Al–4V alloy was characterized by a greater tendency to a decrease in hardness at temperatures arising in the cutting zone. As a result, in machining Ti-555 alloy, the cutting force values are 20% higher than those in machining Ti–6Al–4V alloy. In turn, in [7], it was shown that in machining the Ti–6Al–4V alloy, an increase in the cutting speed up to ultrahigh values (up to 4400 m/min) resulted in a significant decrease in the cutting forces. In [8], authors considered milling of Ti–6Al–4V alloy with an uncoated tool and a tool with the TiN–(Ti,Al)N coating. For the uncoated tool, the temperature on the flank face did not change with an increase in the cutting speed, and there were no signs of obvious abrasive or fatigue wear. Meanwhile, for the tool with the TiN–(Ti,Al)N coating, there was an increase in temperature on the flank face with an increase in the cutting speed, and the significant abrasive wear was also observed. Thus, the coating is able not only to improve, but also to deteriorate the tool properties in milling of titanium alloys. In [9], it was found that the feed and depth of cutting produced a greater effect on the values of the cutting forces, while the cutting speed produced a greater effect on the temperature in the cutting zone. In [10], authors considered the effect produced by the flank wear on the temperature in the cutting zone in milling of workpieces made of Ti–6Al–4V alloy with carbide milling cutters with the TiN coating. As a result of the simulation conducted in combination with the cutting tests, it was found that the feed and the cutting speed influence the temperature in the cutting zone. At the same time, the temperature grows with an increase in the feed and falls with an increase in the cutting speed. The studies revealed that the values of the cutting forces were influenced more by the feed and less by the cutting speed. The influence of the silicon content in the titanium allow on its machinability was considered in [11]. It was found that the presence of silicon significantly increased the machinability of titanium alloys and thus led to a noticeable decrease in the cutting force. Meanwhile, an increase in the silicon content up to 10% and higher resulted in an increase in the cutting force due to an increase in the alloy hardness. The authors argue that with an increase in the feed, the cutting forces grow, and the influence of silicon on a decrease in the cutting forces grows with an increase in the cutting speed. In [12], authors considered the diffusion processes in machining of workpieces made of Ti54M titanium alloy with an uncoated carbide tool of (WC + Co). In particular, it was found that C, W, and Co diffused from the carbide into the material being machined, while the diffusion depths of C and Co were higher than of W. On the border of the tool material and the material being machined, a solid-phase TiC was formed, and it contributed to more active abrasive wear.

One of the most important resources to increase the reliability and efficiency of end milling is the use of cutting tools with wear-resistant coatings. In particular, end milling cutters with multilayered nano-structured (Ti,Al)CN–VCN coating showed a good result in end milling of Al 7010-T7651 alloy and turning Ti–6Al–4V alloy [13]. In [14], authors considered high-speed dry end milling of Ti–6Al–4V titanium alloy with end milling cutters with the (Ti,Al)N–TiN coating. In [15], authors studied end milling of workpieces made of Ti-6242S titanium alloy of aircraft quality with a carbide tool with multilayered TiN–TiC–TiCN coating. It was found that the flank wear prevailed, when the wear value

reached about 0.3 mm, and brittle fracture occurred, together with plastic deformations and interlayer delaminations in the coating structure. For uncoated and coated tools, the key wear mechanisms were adhesive and diffusive wear. The application of coating increased the tool life by two times. In [16], authors considered machining of Ti–6Al–4V titanium alloy with a carbide tool with nano-structured (Al,Ti)N–Si$_3$N$_4$ and (Al,Cr)N–Si$_3$N$_4$ coatings. For the tool with the above coatings, the adhesive wear was the major wear mechanism. Meanwhile, the diffusive and oxidation wear processes also took place for the tool with the (Al,Cr)N–Si$_3$N$_4$ coating. In [17], end milling of Ti–6Al–4V alloy with carbide tools with the (Al,Ti)N and (Ti,Al,Cr)N coatings was considered. The authors noted that the key points in machining of the specified alloy were high temperature and adhesion between the tool material and the material being machined. It was recommended to apply the coatings with a high Al content and a grain size of about 5 nm to ensure the high plasticity and toughness, since these properties were more important than high hardness. In turn, in [18], in machining Ti–6Al–4V alloy, tools with the (Al,Ti)N–WN, (Al,Ti)N-MoN, (Al,Ti)N–CrN, (Al,Ti)N-VN and (Al,Ti)N–NbN were used. The highest tool life parameter and the optimal wear mechanism were registered for a tool with the (Al,Ti)N–VN coating. In [19], authors considered the friction coefficient as the main parameter to influence the tool wear in machining Ti–6Al–4V titanium alloy. It was found that the use of the tool with the (Al,Ti)N coating made it possible to significantly decrease the friction coefficient and the wear of cutting tools.

Thus, for end milling of titanium alloys, tools with the coatings based on titanium nitrides or titanium aluminum are mainly recommended. While the tools with the coatings considered above showed the sufficiently high efficiency in machining titanium alloys, it can be assumed that the application of the coatings with compositions containing no titanium or its compounds can additionally improve the machining performance. Due to the affinity of the chemical composition, it is possible to assume more intensive wear of tools with titanium-containing coatings due to the processes of interdiffusion between the tool material and the material being machined, especially those intensifying at high temperatures typical for the cutting zone [15,20,21]. Another factor which is able to stimulate an increased tool wear is the high adhesion between the material being machined (titanium alloy) and the titanium-containing coating [20,21].

Multilayered nano-structured Zr–ZrN–(Zr,Al)N, Zr–ZrN–(Zr,Cr,Al)N coatings, which showed their high efficiency in machining various materials, were used as coatings for tools in end milling of titanium alloys [22–31]. An important property of the specified coatings is a combination of high hardness (up to 45 GPa [31]) with high thermal resistance and good elastic–plastic characteristics, which is of special importance in conditions of interrupted cutting at high temperatures. The absence of titanium and its compounds in the composition of the specified coatings makes it possible to predict a decrease in diffusion and adhesion processes and the related tool wear. Tools with the commercial TiN and (Ti,Al)N coatings and uncoated tools were used as reference samples.

## 2. Materials and Methods

The coating was deposited at the VIT-2 (MSTU STANKIN-IKTI RAN, Moscow, Russia) unit using the technology of filtered cathodic vacuum arc deposition (FCVAD) [32–39]. The deposition process parameters ($n$—the rotation speed of a turntable, $I_{Zr}$—the Zr cathode arc current, $I_{Al}$—the Al cathode arc current, $I_{Ti}$—the Ti cathode arc current, $I_{Cr}$—the Cr cathode arc current, $U_b$—the value of the displacement potential on the substrate, and $P_N$—the pressure of the reaction gas (nitrogen)) are presented in Table 1. During the deposition process, three-cathode systems with Zr (99.98%), Cr (99.97%), Ti (99.98%) and Al (98.2% + 1% Si) cathodes were used.

The cutting tests were conducted on a WF 4.1 vertical milling machine (Knuth Werkzeugmaschinen GmbH, Hamburg, Germany) (Figure 1).

**Table 1.** Deposition parameters for the coatings under study.

| Architecture of Coating | Parameters for Deposition Process | | | | | | |
|---|---|---|---|---|---|---|---|
| | $n$ (rev/min) | $I_{Zr}$ (A) | $I_{Al}$ (A) | $I_{Ti}$ (A) | $I_{Cr}$ (A) | $U_b$ (V) | $P_N$ (Pa) |
| Ti–TiN | 1.2 | – | – | 70 | – | −160 | 0.4 |
| Ti–TiN–(Ti,Al)N | | – | 160 | 60 | – | −160 | 0.4 |
| Zr–ZrN–(Zr,Al)N | | 55 | 160 | 60 | – | −160 | 0.4 |
| Zr–ZrN–(Zr,Cr,Al)N | | 55 | 170 | 60 | 75 | −160 | 0.4 |

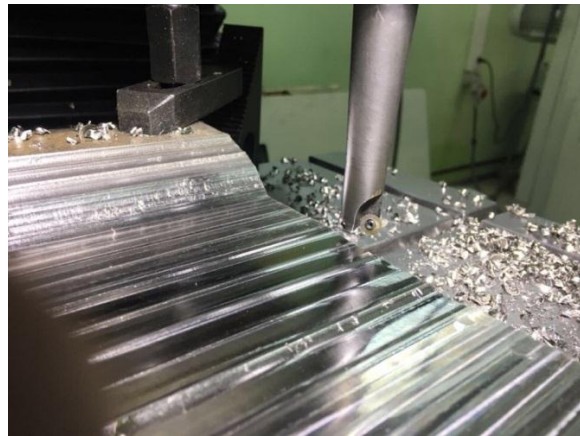

**Figure 1.** Area of research in end milling on a WF 4.1 vertical milling machine.

VT20 (Al: 6%, V: 2%, Zr: 2%, and Mo: 1.5%) titanium alloy used in the aerospace was taken as the material to be machined. The VT20 alloy of the Al–Mo–V–Zr system is a typical alloy with low content of isomorphic p-stabilizers within the range of their solubility in α phase. Residual β phase in it is insignificant. The process of milling was held at the following cutting modes: $f_z = 0.11$ mm/tooth, $a_p = 1$ mm, at cutting speeds of $v_c = 62.8$ m/min, $v_c = 81.2$ m/min and $v_c = 102.1$ m/min. The flank wear land of 0.4 mm was assumed as a wear criterion. In the basic experimental studies, the R300-016B20L-08L end milling cutter with a diameter of 16 mm (Sandvik Coromant, Sweden) was used. The general view of the end milling cutter is presented in Figure 2. The end milling cutter was equipped with replaceable disposable carbide (HC grade) inserts of Coromill 360-0828 (R300-0828Z-PM 1130) (Sandvik Coromant, Sweden) with the developed multilayered composite coatings.

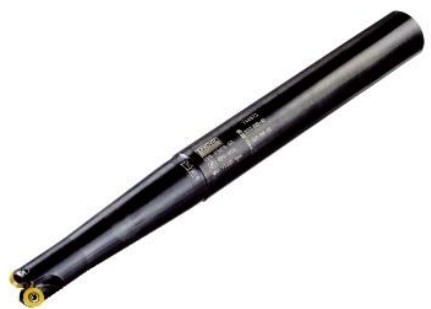

**Figure 2.** General view of Sandvik Coromant R300-016B20L-08L end milling cutter with a diameter of 16 mm.

The microstructure of the surfaces and cross sections of the obtained samples were studied using an FEI Quanta 600 FEG scanning electron microscope (SEM) (Materials & Structural Analysis Division, Hillsboro, OR, USA) with a field emission cathode and an integrated attachment of an EDAX energy dispersive X-ray microanalyzer.

The analysis of the elemental compositions of the samples was studied using an energy-dispersive X-ray spectroscopy (EDXS; INCA Energy; Oxford Instruments, Abingdon, UK).

The adhesion strength of the coatings was studied using a Nanovea scratch tester (Micro Scratch, Nanovea, Irvine, CA, USA) under the standard of ASTM C1624-05 [40]. The hardness (HV) of the coatings was determined by measuring the indentation at low loads according to the method by Oliver and Pharr [41], which was carried out on a micro-indentometer microhardness tester (CSM Instruments) at a fixed load of 30 mN.

## 3. Results and Discussion

The conducted microstructure studies of the specified coatings (Figure 3) detected the typical columnar structure in the TiN coating. The Ti–TiN–(Ti,Al)N, Zr–ZrN–(Zr,Al)N, and Zr–ZrN–(Zr,Cr,Al)N coatings have nano-structured wear-resistant layers with average thicknesses of the binary nanolayer of λ = 150, 120, and 80 nm, respectively. The total thickness of the coatings is identical and is equal to about 3.5 μm (taking into account the uniformity of the coating thickness).

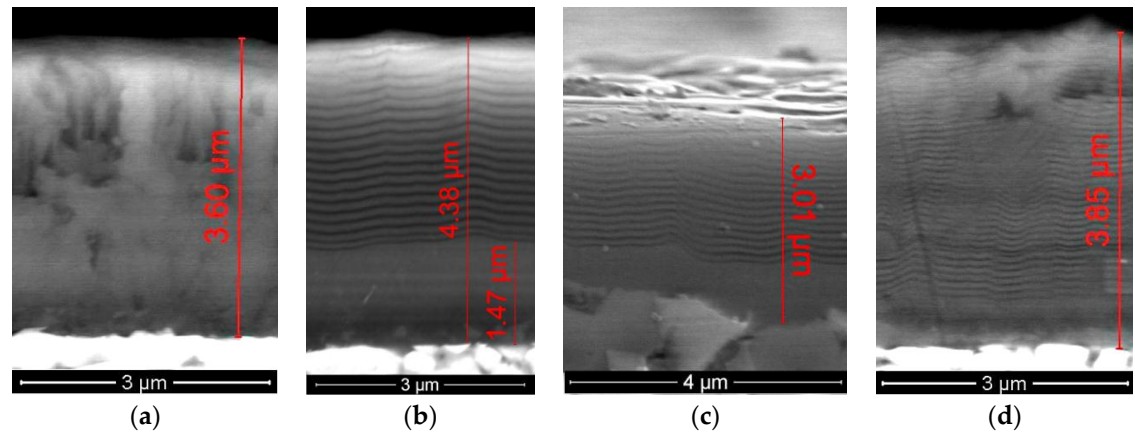

**Figure 3.** Microstructures of the coatings under study (SEM): (**a**) Ti–TiN, (**b**) Ti–TiN–(Ti,Al)N, (**c**) Zr–ZrN–(Zr,Al)N, and (**d**) Zr–ZrN–(Zr,Cr,Al)N.

The chemical analysis of the coatings detected compliance of their nominal and investigated elemental compositions (Figure 4).

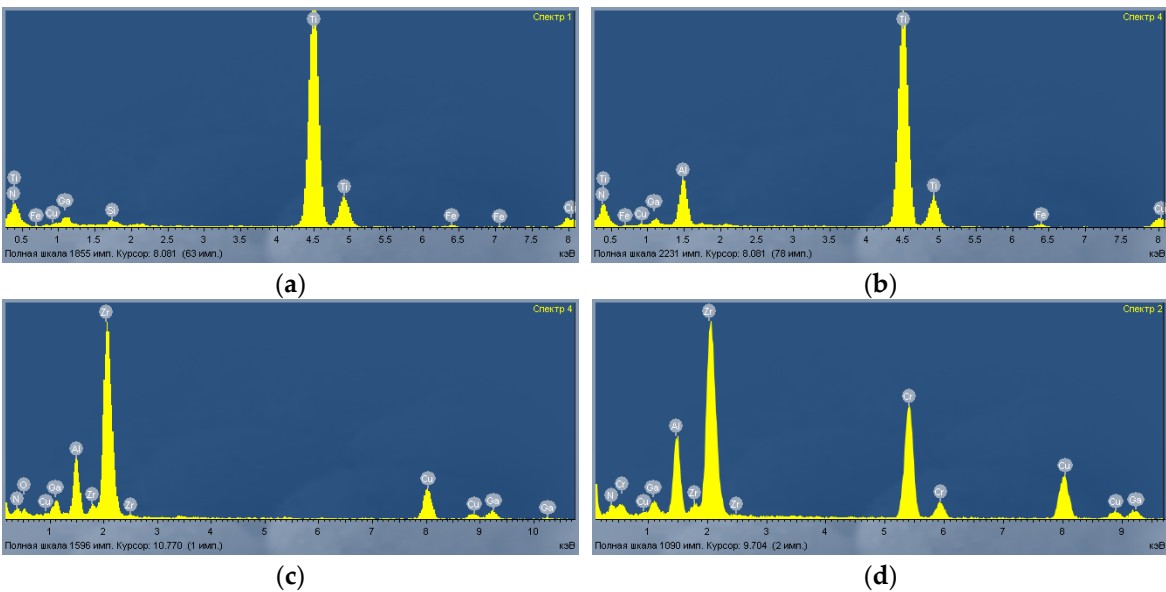

**Figure 4.** Results for the analyses of the elemental compositions of the coatings under study (wear-resistant layer): (**a**) Ti–TiN, (**b**) Ti–TiN–(Ti,Al)N, (**c**) Zr–ZrN–(Zr,Al)N, and (**d**) Zr–ZrN–(Zr,Cr,Al)N.

The data on the microhardness and critical failure force $L_{C2}$ are presented in Table 2. All coatings under study are characterized by sufficiently high hardness and good resistance to fracture in scratch testing. The hardness of the Zr–ZrN–(Zr,Cr,Al)N coating corresponds to the data [31] for a similar nitrogen pressure. The higher hardness of the Zr–ZrN–(Zr,Cr,Al)N coating compared to the Zr–ZrN–(Zr,Al)N coating can be explained by a smaller grain size, the growth of which is constrained by the introduction of Cr [42]. The introduction of Al and Cr distorting the nitride lattice, due to its lower radius, improved resistance to plastic deformation (solid solution hardening) [42].

**Table 2.** Results for the study of the mechanical properties of the coatings.

| Coating Type | Microhardness (HV) (GPa) | Critical Failure Force $L_{C2}$ (N) |
|---|---|---|
| Ti–TiN | 27.1 | 32.7 |
| Ti–TiN–(Ti,Al)N | 32.3 | 38.4 |
| Zr–ZrN–(Zr,Al)N | 30.2 | >40 |
| Zr–ZrN–(Zr,Cr,Al)N | 34.7 | >40 |

The results of the cutting tests at the cutting speeds of 62.8 m/min, 81.2 m/min and 102.1 m/min are presented in Figure 5.

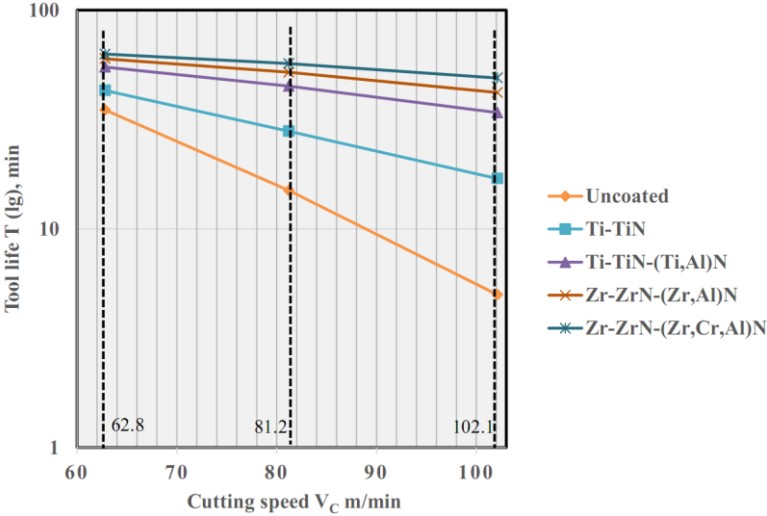

**Figure 5.** Dependence of tool life on the cutting speed in end milling of VT20 titanium alloy with tools with the coatings under study ($f_z$ = 0.11 mm/tooth, and $a_p$ = 1 mm).

After examining the results of the cutting tests, it is possible to conclude that with an increase in the cutting speed for an uncoated tool, its tool life falls. Meanwhile, at the cutting speed of $v_c$ = 102.1 m/min, the use of an uncoated tool becomes unreasonable, since its tool life is extremely short. For the tool with the Ti–TiN coating, the tool life parameter decreases less intensively. However, at the cutting speed of $v_c$ = 102.1 m/min, the tool operation also becomes unreasonable because of its short tool life. Tools with nano-structured coatings showed fairly close tool life values, with the difference increasing with an increase in the cutting speed. The lowest decrease in the tool life with an increase in the cutting speed was shown by the tool with the Zr–ZrN–(Zr,Cr,Al)N coating, and this fact can be explained by good thermal and shock resistance of the above coating. For the tool with the Ti–TiN–(Ti,Al)N coating, the tool life parameter decreases with greater intensity, and this can be explained by its lower thermal and brittle fracture resistance. With the tool life parameter, the tool with the Zr–ZrN–(Zr,Al)N coating is on the intermediate position between the two tools with the nano-structured coatings considered above. Let us consider the wear process for a tool at an "average" cutting speed of $v_c$ = 81.2 m/min. The wear mechanism for a carbide tool can be explained by parallel wear mechanisms, including abrasion, adhesive-fatigue, oxidation and diffusion processes [1].

The results of the wear process after 12 min of milling are presented in Figure 6. Following the obtained results, it can be concluded that multilayered composite nano-structured coatings provide a significant decrease in the development of the wear centers on the flank and rake faces. The study of the failure dynamics for carbide inserts in end milling of titanium alloy found that when an uncoated tool and a tool with the Ti–TiN coating were used and when the criterion of the major flank wear was reached, clear brittle fracture and chipping took place. Meanwhile, when a tool with multilayered composite nano-structured coating was used, the chipping and spallation were less clear. For the tool with the Ti–TiN–(Ti,Al)N coating, a crater was formed on the rake face, while no formation of such a crater was observed for tools with other types of coatings and an uncoated tool.

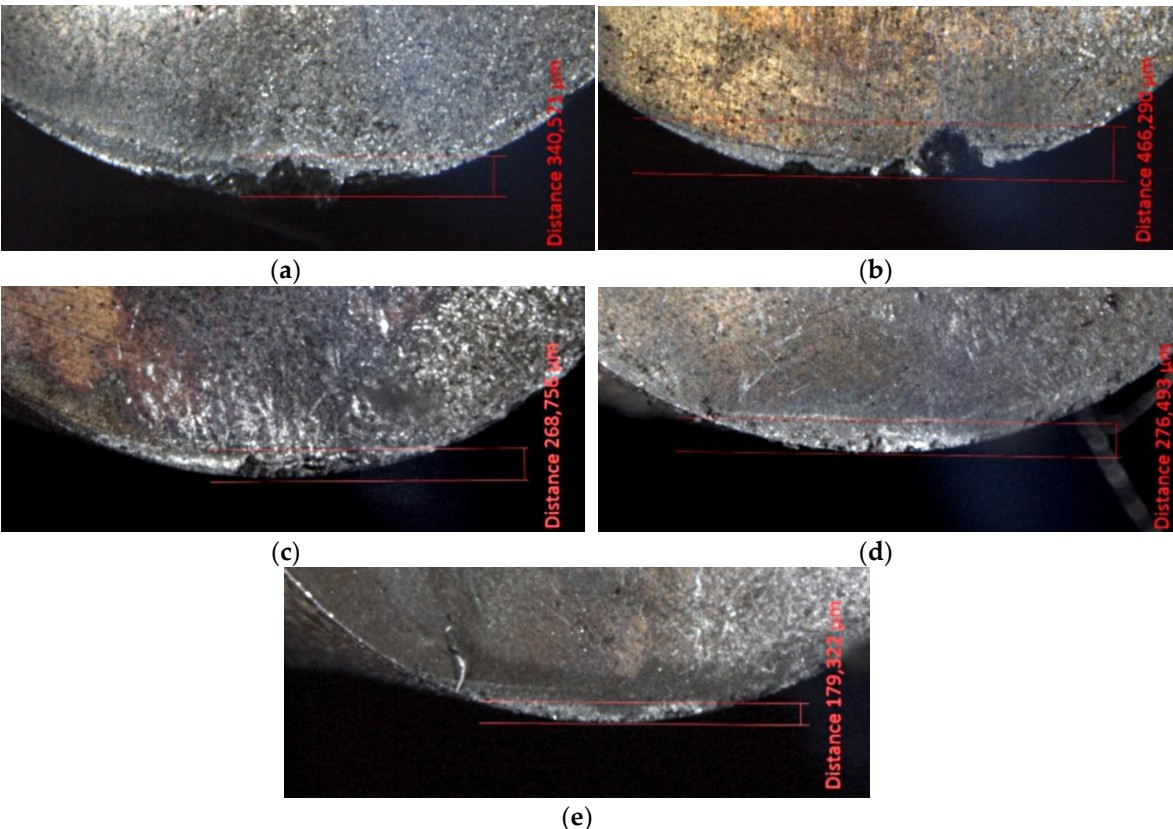

**Figure 6.** Wear patterns for contact pads of carbide inserts: uncoated (**a**) and with coatings Ti–TiN (**b**), Ti–TiN–(Ti,Al)N (**c**), Zr–ZrN–(Zr,Al)N (**d**), and Zr–ZrN–(Zr,Cr,Al)N (**e**) after 12 min of milling at $v_c = 81.2$ m/min, $f_z = 0.11$ mm/tooth, and $a_p = 1$ mm.

Let us consider in more detail the wear process for the tool under study on cross sections made perpendicular to the cutting edge (Figure 7). An extensive adherent of the material being machined is formed on the flank and rake faces of the uncoated tool (Figure 7a), and there is also an area of plastic deformation of the tool material formed on the rake face. There is a noticeable area of brittle fracture of the cutting edge on the tool with the Ti–TiN coating (Figure 7b), and there is also an extensive adherent of the material being machined. For the tool with the Ti–TiN–(Ti,Al)N coating (Figure 7c), there is a crater on the rake face and signs of brittle fracture on the flank face. An extensive adherent of the material being machined is formed in a crater and on the flank face of the tool. For the tool with the Zr–ZrN–(Zr,Al)N coating, there is a noticeable plastic deformation of the tool material on the rake face, and there is an adherent of the material being machined on the flank face. Finally, for the tool with the Zr–ZrN–(Zr,Cr,Al)N coating, there is no formation of any significant adherents of the material being machined; in general, the rake face retains its shape and there are insignificant signs of brittle fracture (chipping) on the flank face.

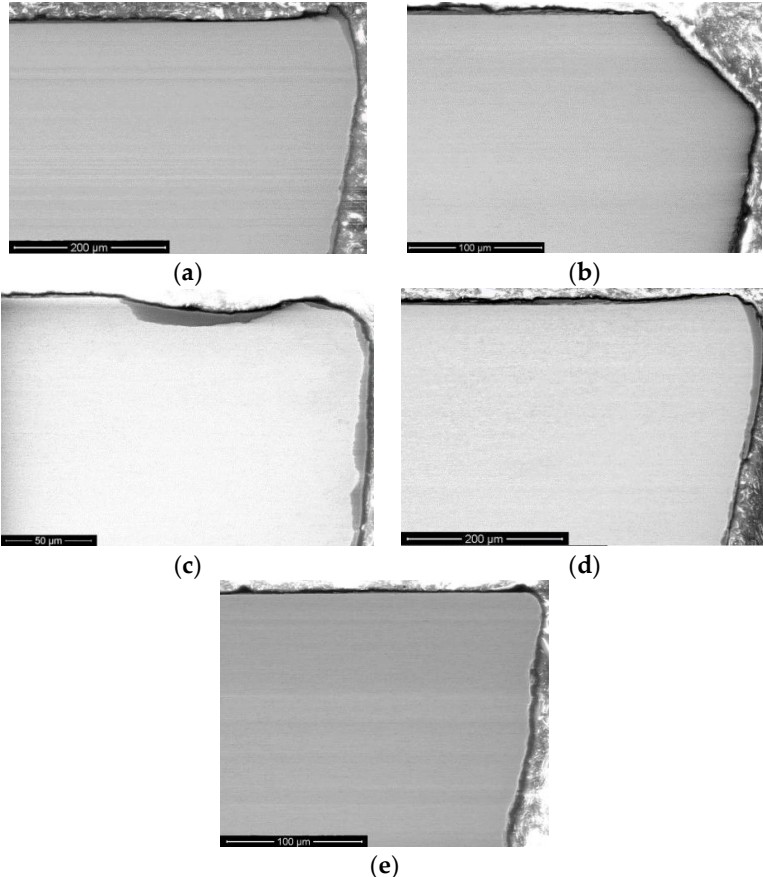

**Figure 7.** Wear patterns for the tool under study on cross sections made perpendicular to the cutting edge: uncoated (**a**) and with coatings Ti–TiN (**b**), Ti–TiN–(Ti,Al)N (**c**), Zr–ZrN–(Zr,Al)N (**d**), and Zr–ZrN–(Zr,Cr,Al)N (**e**) after 12 min of milling at $v_c$ = 81.2 m/min, $f_z$ = 0.11 mm/tooth, and $a_p$ = 1 mm.

Let us consider in more detail the wear processes for each of the tools under study. For the uncoated tool, the active abrasive wear and chipping on the flank face are typical (Figure 8a), as well as plastic deformations and chipping on the flank face (Figure 8b). There is a system of branching cracks in the structure of the adherent of the material being machined on the flank face (Figure 9), and this may indicate the presence in the area of cyclic thermal stresses and the high temperature close to the melting point of titanium (1670 °C).

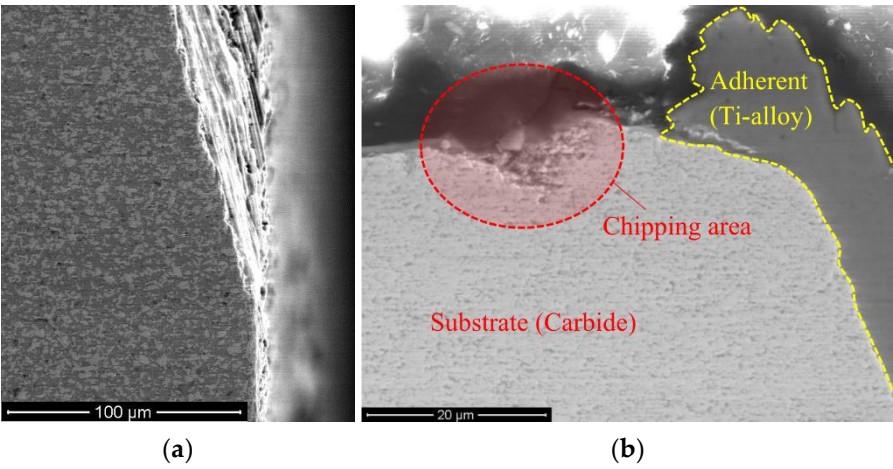

**Figure 8.** Wear pattern for the uncoated tool after 12 min of milling at $v_c$ = 81.2 m/min, $f_z$ = 0.11 mm/tooth, and $a_p$ = 1 mm at the flank face (**a**) and rake face (**b**).

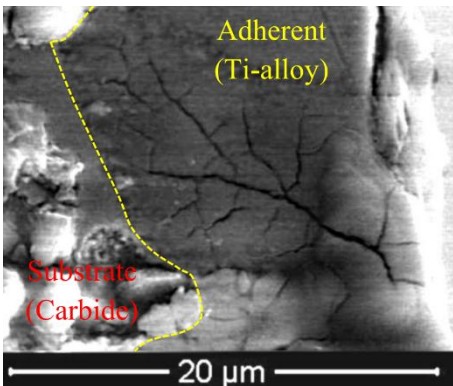

**Figure 9.** Formation of microcracks in the adherent structure of the material being machined on the flank face of the uncoated tool.

During the consideration of the wear process for the tool with the Ti–TiN coating (Figure 10), it is possible to notice the formation of an extensive adherent of the material being machined in the area of the brittle fracture of the tool (Figure10a). When studying the wear process on the coating (Figure 10b,c), it is possible to notice signs of active brittle fracture, accompanied by tearing off fragments of the coating. There are extensive longitudinal cracks, delaminations in the coating structure, and this may indicate the prevailing mechanism of the adhesive-fatigue failure.

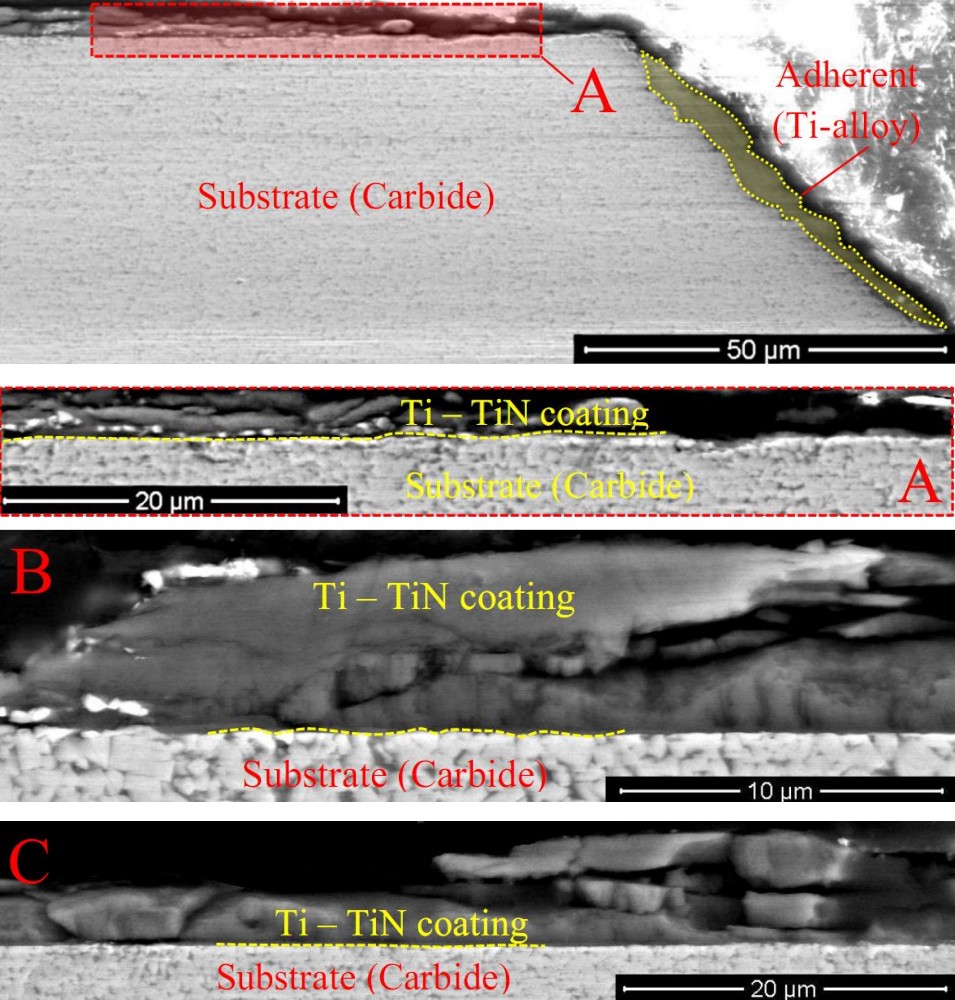

**Figure 10.** Wear pattern for the tool with Ti–TiN coating.

For the tool with the Ti–TiN–(Ti,Al)N coating (Figure 11), the wear mechanism is characterized by a combination of both adhesive-fatigue and diffusive and abrasive processes. The tool with the specified coating is the only one with the registered formation of a noticeable crater on the rake face. Meanwhile, the wear mechanism of the Ti–TiN–(Ti,Al)N coating itself is close to the wear mechanism for the Ti–TiN coating considered above. Despite the noticeable difference in the structure of the considered coatings (monolithic and nano-structured), the Ti–TiN–(Ti,Al)N coating is also characterized by the formation of extensive longitudinal cracks and delaminations in combination with torn-off fragments of the coating (Figure 11a,d). In the area, immediately adjacent to the cutting tip, it is possible to register the sharp, "breaking" coating failure (Figure 11b). When studying the nature of crack propagation in the coating structure, it is possible to note that the crack does not only grow along the boundaries between the nanolayers, but also breaks the structure of nanolayers (Figure 11c). It can be argued that for the given coating in the specified conditions of cutting, the mechanism of brittle fracture with adhesive processes is typical.

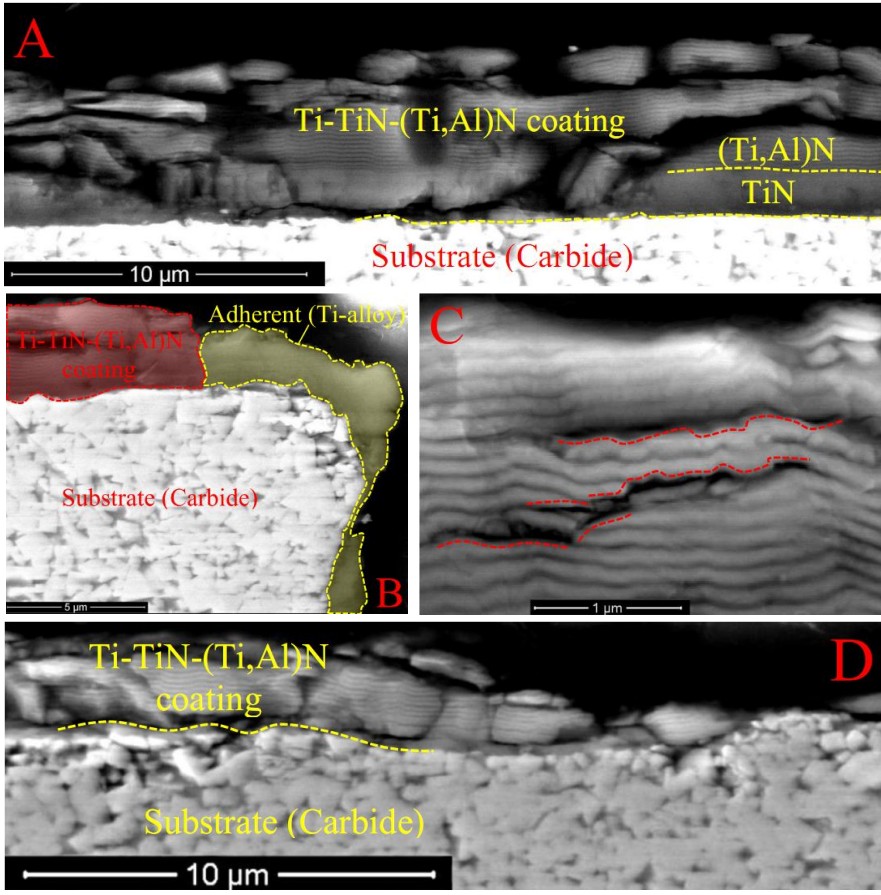

**Figure 11.** Wear pattern for the tool with Ti–TiN–(Ti,Al)N coating.

The wear mechanism of the Zr–ZrN–(Zr,Al)N coating has a number of specific features compared to the above (Figure 12). It can be argued that this coating is more plastic, and it is better retained on the rake face of the tool. Given that the adhesive processes continue to play a certain role in the wear of the tool with the coating under study, it can be argued about a decrease in their influence. In particular, despite the formation of delaminations in the coating structure, this does not lead to its significant or complete failure (Figure 12c,d). In some cases, fragments of the coating are torn off, but the process is much more localized (Figure 12b). During consideration of the area immediately adjacent to the cutting edge (Figure 12a), it is possible to notice signs of abrasive wear in the carbide structure. Meanwhile, the wear process is much more balanced, without noticeable brittle fracture and spallation from the substrate.

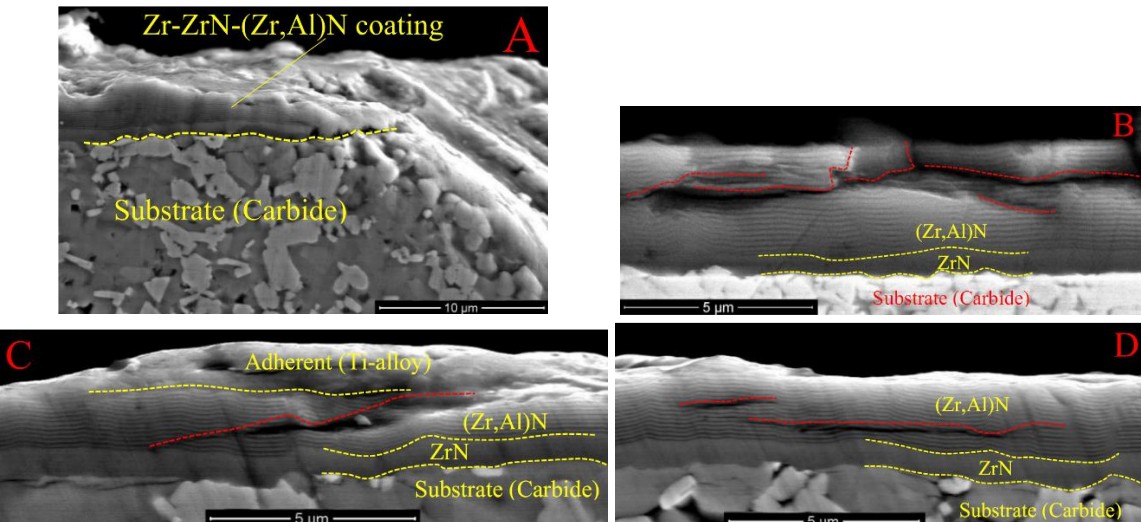

**Figure 12.** Wear pattern for the tool with Zr–ZrN–(Zr,Al)N coating.

Finally, the tool with the Zr–ZrN–(Zr,Cr,Al)N coating showed the most balanced wear process (Figure 13). The coating retains its integrity to the greatest extent on the rake face of the tool. The coating failure on the flank face is directly limited by the flank wear land. There are only insignificant delaminations in the coating structure, and they indicate the effect of the adhesive-fatigue processes. There are also minor fragments torn off the surface and bounded by the boundaries of the nanolayers.

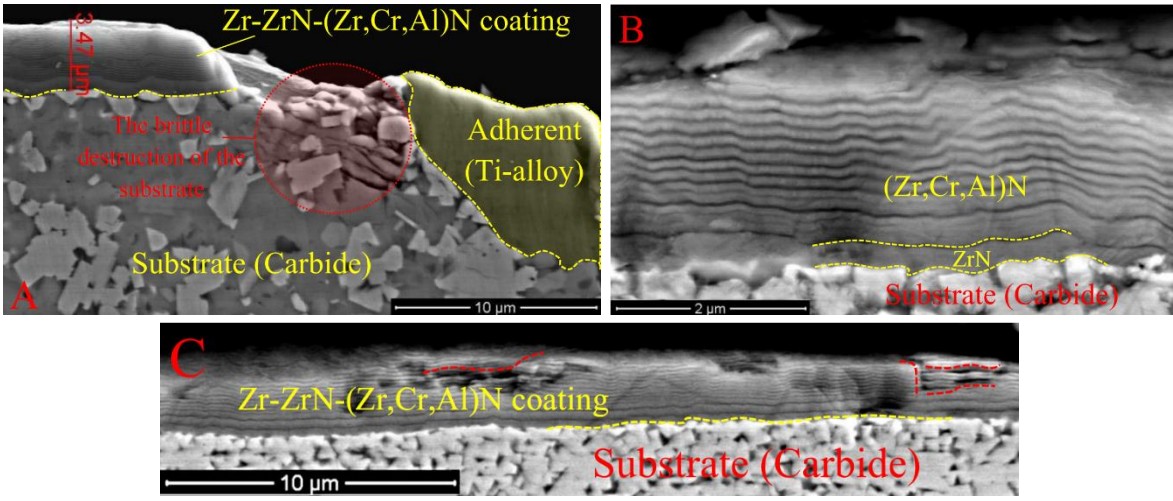

**Figure 13.** Wear pattern for the tool with the Zr–ZrN–(Zr,Cr,Al)N coating.

## 4. Conclusions

The objective of the study was to improve the efficiency of end milling of titanium alloys by using a tool with multilayered composite coating with nano-structured wear-resistant layer based on zirconium nitride. The tools with the Zr–ZrN–(Zr,Al)N and Zr–ZrN–(Zr,Cr,Al)N coatings were considered. The tools with the multilayered composite nano-structured Ti–TiN–(Ti,Al)N coating, the commercial Ti–TiN coating, and the uncoated tool were considered as objects of comparison. The conducted studies found that all the examined coatings were characterized by the considerable high microhardness (over 27 GPa) and the critical failure force $L_{C2}$ (over 30 N). Meanwhile, the highest microhardness of 34.7 GPa was registered for the Zr–ZrN–(Zr,Cr,Al)N coating. The cutting tests conducted at the cutting speeds of 62.8 m/min, 81.2 m/min and 102.1 m/min showed that the use of the tool with multilayered composite nano-structured coating increased the tool life by

2–3 times as compared to the uncoated tool and by 1.5–2 times as compared to the tool with the Ti–TiN coating. Meanwhile, the longest tool life was registered for the tool with the Zr–ZrN–(Zr,Cr,Al)N coating, and the tool showed the smallest fall in the tool life with an increase in the cutting speed. The conducted studies of the wear processes typical for the tools with the coatings under study and for the uncoated tools showed a number of significant differences. In particular, for the tools with the Ti–TiN and Ti–TiN–(Ti,Al)N coatings, the brittle fracture of the coating on the rake face and extensive delaminations with large torn-off fragments of the coatings were typical, and this can indicate intensive adhesive-fatigue and interdiffusion processes. For the tools with the Zr–ZrN–(Zr,Al)N and Zr–ZrN–(Zr,Cr,Al)N coatings, the significantly more balanced wear mechanism is typical, the coating on the rake face in general retains its integrity, and abrasive wear is more clear. It can be concluded that the multilayered composite nano-structured coatings based on zirconium nitride can be efficiently applied as wear-resistant coatings in end milling of titanium alloys.

**Author Contributions:** Conceptualization, A.V. and M.O.; Methodology, A.V., M.O., N.S. and B.M.; Validation, K.D. and V.P.; Formal Analysis, M.O.; Investigation, M.O., Y.B., N.S., K.D. and V.P.; Resources, A.V.; Data Curation, M.O.; Writing-Original Draft Preparation, A.V. and M.O.; Writing-Review & Editing, A.V. and M.O.; Visualization, M.O.; Supervision, A.V.; Project Administration, A.V.; Funding Acquisition, A.V.

**Funding:** This study was funded by a grant of the Russian Science Foundation (theme No. 18-24/RNF Agreement No. 18-19-00312 dated April 20, 2018).

**Conflicts of Interest:** The authors declare no conflicts of interest.

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
