# Peer review of "Increase in Efficiency of End Milling of Titanium Alloys Due to Tools with Multilayered Composite Nano-Structured Zr-ZrN-(Zr,Al)N and Zr-ZrN-(Zr,Cr,Al)N Coatings"

_coatings, doi:10.3390/coatings8110395_

Round 1
Reviewer 1 Report
This study has mainly explored the effect of the use of Zr-ZrN-(Zr,Al)N and Zr-ZrN-(Zr,Cr,Al)N coatings on the tool life after end milling titanium alloy. The key point in this study is understandable, but some puzzled and missing contents need to be further confirmed as below:
1. What is the composition of the tool steel used as the substrate in the study?
2. The conditions of coating process are not stated in “Materials and Methods” such as target, temperature, current, and working pressure.
3. The captions in some figures are not listed such as Figures 7a–e and 8a,b. Also, the image of Figure 5 is not clear.
4. In Figure 10, there are two “A” symbols, while no “d” symbol. However, Figure 10b–d was written in the text (line 237).
5. EDXS was used to study chemical composition of the samples (line 153–154). However, as we known, the accuracy of composition analysis maybe has a controversial question because EDXS only has the function of semi-quantitative analysis.
6. Overall, in my opinion, the article in the current situation is not suitable for the publication in journal of coatings.
Author Response
1. What is the composition of the tool steel used as the substrate in the study?
Substrate-cemented carbide (Grade-HC by Sandvik Coromant nomenclature) (added in the text)
2. The conditions of coating process are not stated in “Materials and Methods” such as target, temperature, current, and working pressure.
Process parameters added (Table 1)
3. The captions in some figures are not listed such as Figures 7a–e and 8a,b. Also, the image of Figure 5 is not clear.
Captions have been added and modified; Figure 5 has been modified.
4. In Figure 10, there are two “A” symbols, while no “d” symbol. However, Figure 10b–d was written in the text (line 237).
The “A” in the upper image denotes the highlighted area, which is then shown below with high magnification. Made changes to the Figure for a better understanding.
Fixed on (Figure 10b,c).
5. EDXS was used to study chemical composition of the samples (line 153–154). However, as we known, the accuracy of composition analysis maybe has a controversial question because EDXS only has the function of semi-quantitative analysis.
Of course, I agree with the reviewer. In this case, the task of studying the chemical composition of coatings was not the main one; it was important to confirm the presence in the composition of all the declared components. This method does not allow to give an exact quantitative value, but allows us to determine the presence of elements in the composition and the qualitative ratio (proportion) of their content.
The wording has been changed and now it seems to me that it has become more correct.
6. Overall, in my opinion, the article in the current situation is not suitable for the publication in journal of coatings.
I hope that after the changes to the article, it has become more acceptable for publication. Thank reviewers!
Reviewer 2 Report
This article investigates an increase in the tool life parameter for metal-cutting tools and efficiency of end milling for titanium alloys due to the use of tools with multilayered composites including nano-structured Zr-ZrN-(Zr,Al)N and Zr-ZrN-(Zr,Cr,Al)N coatings. Direct comparison among tool lifetimes of the composites were conducted. This article is well written and is useful. Thus, it can be accepted.
Author Response
This article investigates an increase in the tool life parameter for metal-cutting tools and efficiency of end milling for titanium alloys due to the use of tools with multilayered composites including nano-structured Zr-ZrN-(Zr,Al)N and Zr-ZrN-(Zr,Cr,Al)N coatings. Direct comparison among tool lifetimes of the composites were conducted. This article is well written and is useful. Thus, it can be accepted.
Thank you!
Reviewer 3 Report
The authors studied the roles of multilayered composite nano-structured Zr-ZrN-(Zr,Al)N and Zr-ZrN-5 (Zr,Cr,Al)N coatings during end milling of Ti-6Al-4V. The manuscript is well written, however, the following points must be considered before the final acceptance for publications:
1. In two coatings, Zr-ZrN-(Zr,Al)N and Zr-ZrN-5 (Zr,Cr,Al)N, the only difference is Cr contents, therefore, the role of Cr in coatings should be explained in abstract shortly.
2. Introduction is well written in this manuscript because the tool wear mechanisms are explained in details with the shed of existing literature. However, the reviewer suggest to include few sentences on the transfer of titanium to the tool surface or built-up edge formation (as this is another common problem in titanium machining as authors found and stated in Figures 8b and 9) from literature such as
-Surface & Coatings Technology 2006, 200, 5663–5676;
-Journal of Materials Processing Technology 2000, 99, 266–274;
-Surface & Coatings Technology 2014, 260, 290–302.
3. Figure 1a is not necessary for the manuscript.
4. It seems like the addition of Cr in Zr-ZrN-(Zr,Al)N increases the hardness from 30.2 to 34.7
(Table 1). The plausible mechanisms should be explained.
5. What is the significance of addition of Zr-ZrN-(Zr,Al, Nb)N results in Figure 5? Why Nb?
6. The authors provided details of microstructure of the coatings before and after the milling experiments in this manuscript, however, the lack of microstructure of Ti-6Al-4V workpiece (to understand the surface integrity) after milling is the major deficiency of this work.
Author Response
1. In two coatings, Zr-ZrN-(Zr,Al)N and Zr-ZrN-5 (Zr,Cr,Al)N, the only difference is Cr contents, therefore, the role of Cr in coatings should be explained in abstract shortly.
Added to abstract
2. Introduction is well written in this manuscript because the tool wear mechanisms are explained in details with the shed of existing literature. However, the reviewer suggest to include few sentences on the transfer of titanium to the tool surface or built-up edge formation (as this is another common problem in titanium machining as authors found and stated in Figures 8b and 9) from literature such as
-Surface & Coatings Technology 2006, 200, 5663–5676;
-Journal of Materials Processing Technology 2000, 99, 266–274;
-Surface & Coatings Technology 2014, 260, 290–302.
Many thanks for the recommendation. I added the data, taking into account the articles recommended by the reviewer.
3. Figure 1a is not necessary for the manuscript.
I agree. Figure 1a has been removed.
4. It seems like the addition of Cr in Zr-ZrN-(Zr,Al)N increases the hardness from 30.2 to 34.7
(Table 1). The plausible mechanisms should be explained.
Added explanation for greater hardness—based on papers: Surf. Coat. Technol. 2012, 206, 3764–3771, and Surf. Coat. Technol. 2007, 201, 5547–5551.
5. What is the significance of addition of Zr-ZrN-(Zr,Al, Nb)N results in Figure 5? Why Nb?
This, of course, is a typo. Must be chrome, not niobium. Figure modified.
6. The authors provided details of microstructure of the coatings before and after the milling experiments in this manuscript, however, the lack of microstructure of Ti-6Al-4V workpiece (to understand the surface integrity) after milling is the major deficiency of this work.
Unfortunately, a detailed study of the quality of the surface was not conducted. Only a check of the roughness of the treated surface with the existing standards was carried out. Such work is planned to be done at the next stage of research. It is also planned to expand the range of coatings (using also coatings with niobium and molybdenum), as well as to investigate the change in cutting forces.
I thank the reviewer for helping to improve the article.
Round 2
Reviewer 1 Report
The authors have responded appropriately and done some work to improve the paper. Thereby, I can agree with the revisions to my prior comments.
Reviewer 3 Report
The authors have responded the reviewer’s quarries satisfactorily. The manuscript can be accepted.